# Linear Decision Rule as Aspiration for Simple Decision Heuristics

**Özgür Şimşek**
Center for Adaptive Behavior and Cognition
Max Planck Institute for Human Development
Lentzeallee 94, 14195 Berlin, Germany
`ozgur@mpib-berlin.mpg.de`

## Abstract

Several attempts to understand the success of simple decision heuristics have examined heuristics as an approximation to a linear decision rule. This research has identified three environmental structures that aid heuristics: dominance, cumulative dominance, and noncompensatoriness. This paper develops these ideas further and examines their empirical relevance in 51 natural environments. The results show that all three structures are prevalent, making it possible for simple rules to reach, and occasionally exceed, the accuracy of the linear decision rule, using less information and less computation.

## 1 Introduction

The *comparison* problem asks which of a number of objects has a higher value on an unobserved criterion. Typically, some attributes of the objects are available as input to the decision. An example is which of two houses that are currently for sale will have a higher return on investment ten years from now, given the location, age, lot size, and total living space of each house.

The importance of comparison for intelligent behavior cannot be overstated. Much of human and animal behavior consists of choosing one object—from among a number of available alternatives—to act on, with respect to some criterion whose value is unobserved at the time. Examples include a venture capitalist choosing a company to invest in, a scientist choosing a conference to submit a paper to, a female tree frog deciding who to mate with, and an ant colony choosing a nest area to live in.

This paper focuses on paired comparison, in which there are exactly two objects to choose from, and its solution using linear estimation. Specifically, it is concerned with the environmental structures that make it possible to mimic the decisions of the linear estimator *using less information and less computation*, asking two questions: How much of the linear estimator do we need to know to mimic its decisions, and under what conditions? How prevalent are these conditions in natural environments? In the following sections, I review several ideas from the literature, develop them further, and investigate their empirical relevance.

## 2 Background

A standard approach to the comparison problem is to estimate the criterion as a function of the attributes of the object, typically as a linear function:

$$\hat{y} = w_0 + w_1 x_1 + w_2 x_2 + ... + w_k x_k, \tag{1}$$

where $\hat{y}$ is the estimate of the criterion, $w_0$ is the intercept, $w_1..w_k$ are the weights, and $x_1..x_k$ are the attribute values. This estimate leads to a decision between objects A and B as follows, where

$\Delta x_i$ is used to denote the difference in attribute values between the two objects:

$$\begin{aligned}
\hat{y}_A - \hat{y}_B &= w_1(x_{1A} - x_{1B}) + w_2(x_{2A} - x_{2B}) + ... + w_k(x_{kA} - x_{kB}) \\
&= w_1\Delta x_1 + w_2\Delta x_2 + ... + w_k\Delta x_k
\end{aligned} \tag{2}$$

$$\text{Decision rule} \quad : \quad \begin{cases} \text{Choose object A} & \text{if } w_1\Delta x_1 + ... + w_k\Delta x_k > 0 \\ \text{Choose object B} & \text{if } w_1\Delta x_1 + ... + w_k\Delta x_k < 0 \\ \text{Choose randomly} & \text{if } w_1\Delta x_1 + ... + w_k\Delta x_k = 0 \end{cases} \tag{3}$$

This decision rule does not need the linear estimator in its entirety. The intercept is not used at all. As for the weights, it suffices to know their sign and relative magnitude. For instance, with two attributes weighted $+0.2$ and $+0.1$, it suffices to know that both weights are positive and that the first one is twice as high as the other.

The literature on simple decision heuristics [1, 2] has identified several environmental structures that allow simple rules to make decisions identical to those of the linear decision rule using less information [3]. These are dominance [4], cumulative dominance [5, 6], and noncompensatoriness [7, 8, 9, 10, 11]. I discuss each in turn in the following sections. I refer to attributes also as *cues* and to the signs of the weights as *cue directions*, as in the heuristics literature. An attribute that *discriminates* between two objects is one whose value differs on the two objects. A heuristic that *corresponds* to a particular linear decision rule is one whose cue directions, and cue order if it needs them, are identical to those of the linear decision rule.

The discussion will focus on two successful families of heuristics. The first is *unit weighting* [12, 13, 14, 15, 16, 17], which uses a linear decision rule with weights of $+1$ or $-1$. The second is the family of *lexicographic* heuristics [18, 19], which examine cues one at a time, in a specified order, until a cue is found that discriminates between the objects. The discriminating cue, and that cue only, is used to make the decision. Lexicographic heuristics are an abstraction of the way words are ordered in a dictionary, with respect to the alphabetical order of the letters from left to right.

## 2.1 Dominance

If all terms $w_i\Delta x_i$ in Decision Rule 3 are nonnegative, and at least one of them is positive, then object A *dominates* object B. If all terms $w_i\Delta x_i$ are zero, then objects A and B are *dominance equivalent*. It is easy to see that the linear decision rule chooses the dominant object if there is one. If objects are dominance equivalent, the decision rule chooses randomly.

Dominance is a very strong relationship. When it is present, most decision heuristics choose identically to the linear decision rule if their cue directions match those of the linear rule. These include unit weighting and lexicographic heuristics, with any ordering of the cues.

To check for dominance, it suffices to know the signs of the weights; the magnitudes of the weights are not needed. I occasionally refer to dominance as *simple dominance* to differentiate it from cumulative dominance, which I discuss next.

## 2.2 Cumulative dominance

The linear sum in Equation 2 may be written alternatively as follows:

$$\begin{aligned}
\hat{y}_A - \hat{y}_B &= (w_1 - w_2)\Delta x_1 + (w_2 - w_3)(\Delta x_1 + \Delta x_2) \\
&\quad + (w_3 - w_4)(\Delta x_1 + \Delta x_2 + \Delta x_3) + ... + w_k(\Delta x_1 + .. + \Delta x_k) \\
&= w_1'\Delta x_1' + w_2'\Delta x_2' + w_3'\Delta x_3' + ... + w_k'\Delta x_k',
\end{aligned} \tag{4}$$

where (1) $\Delta x_i' = \sum_{j=1}^{i} \Delta x_j$, $\forall i$ , (2) $w_i' = w_i - w_{i+1}$, $i = 1, 2, .., k-1$, and (3) $w_k' = w_k$.

To this alternative linear sum in Equation 4, we can apply the earlier dominance result, obtaining a new dominance relationship called *cumulative dominance*. Cumulative dominance uses an additional piece of information on the weights: their relative ordering.

Object A *cumulatively dominates* object B if all terms $w_i'\Delta x_i'$ are nonnegative and at least one of them is positive. Objects A and B are *cumulative-dominance equivalent* if all terms $w_i'\Delta x_i'$ are zero. The linear decision rule chooses the cumulative-dominant object if there is one. If objects are

cumulative-dominance equivalent, the linear decision rule chooses randomly. Note that if weights $w_1..w_k$ are positive and decreasing, it suffices to examine $\Delta x_i'$ to check for cumulative dominance (because $w_i' > 0, \ \forall i$).

As an example, consider comparing the value of two piles of US coins. The attributes would be the number of each type of coin in the pile, and the weights would be the financial value of each type of coin. A pile that contains 6 one-dollar coins, 4 fifty-cent coins, and 2 ten-cent coins cumulatively dominates (but not simply dominates) a pile containing 3 one-dollar coins, 5 fifty-cent coins, and 1 ten-cent coin: $6 > 3, \ \ 6 + 4 > 3 + 5, \ \ 6 + 4 + 2 > 3 + 5 + 1$.

Simple dominance implies cumulative dominance. Cumulative dominance is therefore more likely to hold than simple dominance. When a cumulative-dominance relationship holds, the linear decision rule, the corresponding lexicographic decision rule, and the corresponding unit-weighting rule decide identically, with one exception: unit weighting may find a tie where the linear decision rule does not [5].

## 2.3 Noncompensatoriness

Without loss of generality, assume that the weights $w_1, w_2, .., w_k$ are nonnegative, which can be satisfied by inverting the attributes when necessary. Consider the linear decision rule as a sequential process, where the terms $w_i \Delta x_i$ are added one by one, in order of nonincreasing weights. If we were to stop after the first discriminating attribute, would our decision be identical to the one we would make by processing all attributes? Or would the subsequent attributes reverse this early decision?

The answer is no, it is not possible for subsequent attributes to reverse the early decision, if the attributes are binary, taking values of 0 or 1, and the weights satisfy the set of constraints $w_i > \sum_{j=i+1}^{k} w_j, i = 1, 2, .., k - 1$. Such weights are called *noncompensatory*. An example is the sequence $1, 0.5, 0.25, 0.125$.

With binary attributes and noncompensatory weights, the linear decision rule and the corresponding lexicographic decision rule decide identically [7, 8].

This concludes the review of the background material. The contributions of the present paper start in the next section.

## 3    A probabilistic approach to dominance

To choose between two objects, the linear decision rule examines whether $\sum_{i=1}^{k} w_i \Delta x_i$ is above, below, or equal to zero. This comparison can be made with certainty, without knowing the exact values of the weights, if a dominance relationships exists. Here I explore what can be done in the absence of such certainty. For instance, can we identify conditions under which the comparison can be made *with very high probability*? As a motivating example, consider the case where 9 out of 10 attributes favor object A against object B. Although we cannot be certain that the linear decision rule will select object A, that would be a very good bet.

I make the simplifying assumption that $|w_i \Delta x_i|$ are independent, identically distributed samples from the uniform distribution in the interval from 0 to 1. The choice of upper bound of the interval is not consequential because the terms $w_i \Delta x_i$ can be rescaled. Let $p$ and $n$ be the number of positive and negative terms $w_i \Delta x_i$, respectively. Using the normal approximation to the sum of uniform variables, we can approximate $\sum_{i=1}^{k} w_i \Delta x_i$ with the normal distribution with mean $\frac{p-n}{2}$ and variance $\frac{p^2+n^2}{12}$. This yields the following estimate of the probability $P_A$ that the linear decision rule will select object A: $P_A \approx P(X > 0)$, where $X \sim N(\frac{p-n}{2}, \frac{p^2+n^2}{12})$.

**Definition:** Object A *approximately dominates* object B if $P(X > 0) \geq c$, where $c$ is a parameter of the approximation (taking values close to 1) and $X \sim N(\frac{p-n}{2}, \frac{p^2+n^2}{12})$.

A similar analysis applies to cumulative dominance.

# 4 An empirical analysis of relevance

I now turn to the question of whether dominance and noncompensatoriness exist in our environment in any substantial amount. There are two earlier results on the subject. When binary versions of 20 natural datasets were used to train a multiple linear regression model, at least 3 of the 20 models were found to have noncompensatory weights [8].[1] In the same 20 datasets, with a restriction of 5 on the maximum number of attributes, the proportion of object pairs that exhibited simple dominance ranged from 13% to 75% [4].

The present study used 51 natural datasets obtained from a wide variety of sources, including online data repositories, textbooks, research publications, packages for R statistical software, and individual scientists collecting field data. The subjects were diverse, including biology, business, computer science, ecology, economics, education, engineering, environmental science, medicine, political science, psychology, sociology, sports, and transportation. The datasets varied in size, ranging from 12 to 601 objects, corresponding to 66–180,300 distinct paired comparisons. Number of attributes ranged from 3 to 21. The datasets are described in detail in the supplementary material.[2]

I present two sets of results: on the original datasets and on binary versions where numeric attributes were dichotomized by splitting around the median (assigning the median value to the category with fewer objects). I refer to the original datasets as *numeric* datasets but it should be noted that one dataset had only binary attributes and many datasets had at least one binary attribute. Categorical attributes were recoded into binary attributes, one for each category, indicating membership in the category. Objects with missing criterion values were excluded from the analysis. Missing attribute values were replaced by means across all objects. A decision was considered to be *accurate* if it selected an object whose criterion value was equal to the maximum of the criterion values of the objects being compared.

Cumulative dominance and noncompensatoriness are sensitive to the units of measurement of the attributes. In this analysis, all attribute values were normalized to lie between 0 and 1, measuring them relative to the smallest and largest values they take in the dataset.

The linear decision rule was obtained using multiple linear regression with elastic net regularization [21], which contains both a ridge penalty and a lasso penalty. For the regularization parameter $\alpha$, which determines the relative proportion of ridge and lasso penalties, the values of 0, 0.2, 0.4, 0.6, 0.8, and 1 were considered. For the regularization parameter $\lambda$, which controls the amount of total penalty, the default search path in the R-language package glmnet [22] was used. Both $\alpha$ and $\lambda$ were selected using cross validation. Specifically, $\alpha$ and $\lambda$ were set to the values that gave the minimum mean cross-validation error in the training set. I refer to the linear decision rule learned in this manner as the *base decision rule*.

On datasets with fewer than 1000 pairs of objects, a separate linear decision rule was learned for every pair of objects, using all other objects as the training set. On larger datasets, the pairs of objects were randomly placed in 1000 folds and a separate model was learned for each fold, training with all objects not contained in that fold. Five replications were done using different random seeds.

**Performance of the base decision rule** The accuracy of the base decision rule differed substantially across datasets, ranging from barely above chance to near-perfect. In numeric datasets, accuracy ranged from 0.56 to 0.98 (mean=0.79). In binary datasets, accuracy was generally lower, ranging from 0.55 to 0.86 (mean=0.74). Compared to standard multiple linear regression, regularization improved accuracy in most datasets, occasionally in large amounts (as much as by 19%). Without regularization, mean accuracy across datasets was lower by 1.17% in numeric datasets and by 0.51% in binary datasets.

**Dominance** Figure 1 shows prevalence of dominance, measured by the proportion of object pairs in which one object dominates the other or the two objects are equivalent. The figure shows four types of dominance in each of the datasets. Simple and cumulative dominance are displayed as

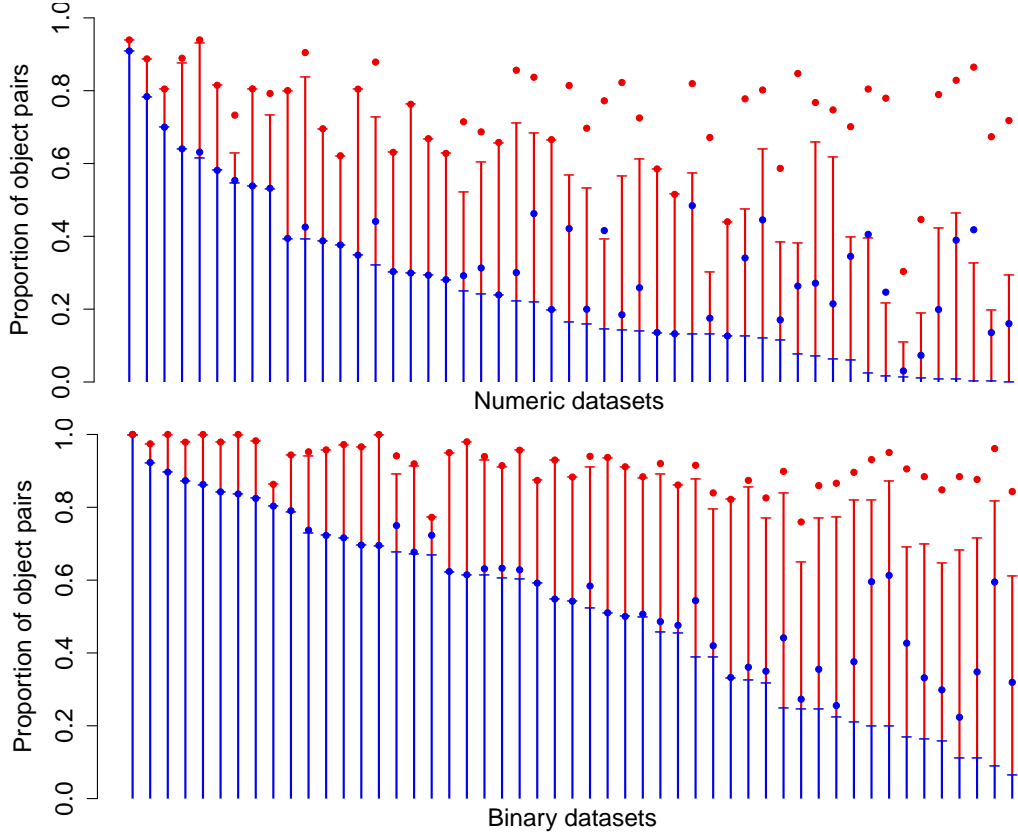

Figure 1: *Prevalence of dominance.* Blue lines show simple dominance, red lines show cumulative dominance, blue-filled circles show approximate simple dominance, and red-filled circles show approximate cumulative dominance.

blue and red lines stacked on top of each other. Recall that simple dominance implies cumulative dominance, so the blue lines show pairs with both simple- and cumulative-dominance relationships. Approximate simple and cumulative dominance are displayed as blue- and red-filled circles, respectively. The datasets are presented in order of decreasing prevalence of simple dominance. The mean, median, minimum, and maximum prevalence of each type of dominance across the datasets

| | NUMERIC DATASETS | | | | BINARY DATASETS | | | |
|---|---|---|---|---|---|---|---|---|
| | Mean | Median | Min | Max | Mean | Median | Min | Max |
| PREVALENCE | | | | | | | | |
| Dom | 0.25 | 0.16 | 0.00 | 0.91 | 0.51 | 0.54 | 0.07 | 1.00 |
| Dom approx c=0.99 | 0.35 | 0.31 | 0.03 | 0.91 | 0.58 | 0.59 | 0.22 | 1.00 |
| Cum dom | 0.58 | 0.62 | 0.11 | 0.94 | 0.87 | 0.89 | 0.61 | 1.00 |
| Cum dom approx c=0.99 | 0.74 | 0.77 | 0.30 | 0.94 | 0.92 | 0.92 | 0.76 | 1.00 |
| Noncompensatory weights | | | | | 0.17 | 0.00 | 0.00 | 1.00 |
| Noncompensation | 0.83 | 0.85 | 0.49 | 0.99 | 0.93 | 0.96 | 0.77 | 1.00 |
| ACCURACY (%) | | | | | | | | |
| Dom | 76.8 | 77.2 | 56.2 | 97.5 | 87.1 | 89.0 | 63.9 | 100.0 |
| Dom approx c=0.99 | 81.2 | 82.9 | 57.0 | 100.5 | 90.5 | 91.2 | 70.6 | 100.0 |
| Cum dom | 90.6 | 93.4 | 60.8 | 101.4 | 98.3 | 98.9 | 90.6 | 103.7 |
| Cum dom approx c=0.99 | 94.2 | 96.1 | 69.5 | 101.4 | 99.2 | 99.6 | 93.8 | 103.7 |
| Lexicographic | 93.5 | 96.1 | 51.4 | 110.6 | 97.6 | 99.6 | 78.9 | 104.4 |

Table 1: Descriptive statistics on dominance, cumulative dominance, and noncompensatoriness. Accuracy is shown as a percentage of the accuracy of the base decision rule.

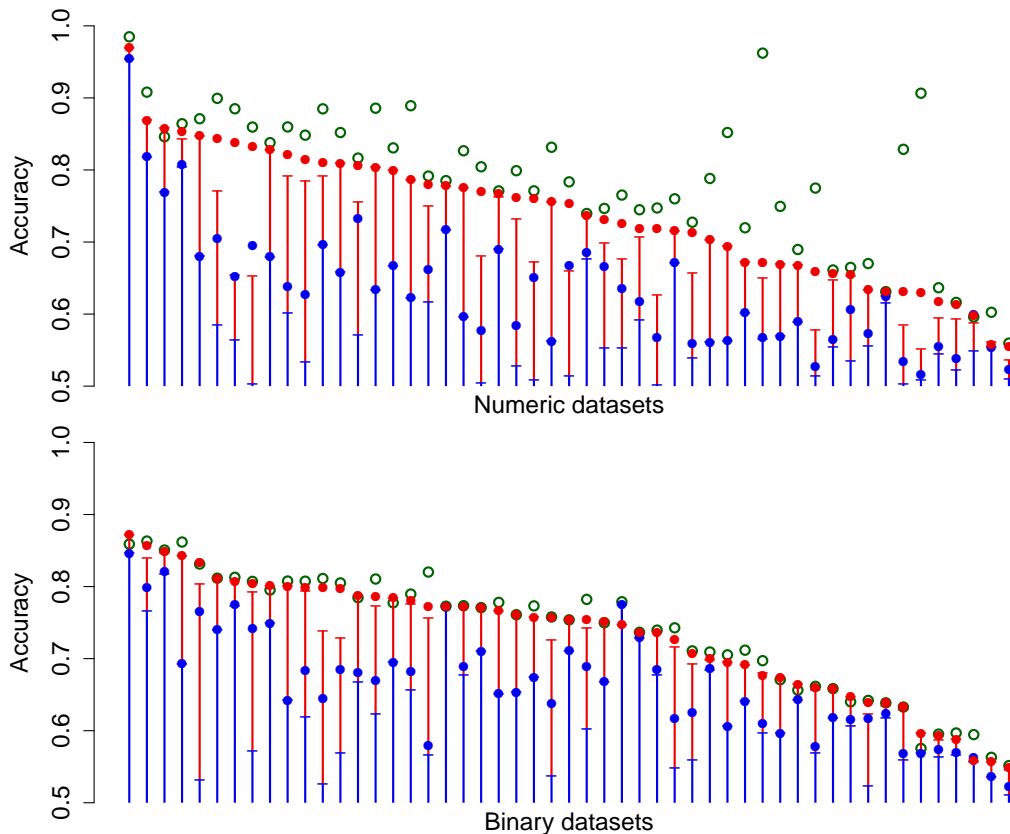

Figure 2: *Accuracy of decisions guided by dominance.* Blue lines show simple dominance, red lines show cumulative dominance, blue-filled circles show approximate simple dominance, and red-filled circles show approximate cumulative dominance. Green circles show the accuracy of the base decision rule for comparison.

are shown in Table 1, along with other performance measures that will be discussed shortly. The approximation made a difference in 27–33 of 51 datasets, depending on type of dominance and data (numeric/binary). As expected, the datasets on which the approximation made a difference were those that had a larger number of attributes. Specifically, they all had six or more attributes.

Figure 2 shows the accuracy of decisions guided by dominance: choose the dominant object when there is one; choose randomly otherwise. This accuracy can be higher than the accuracy of the base decision rule, which happens if choosing randomly is more accurate than the base decision rule on pairs that exhibit no dominance relationship. Table 1 shows the mean, median, minimum, and maximum accuracies across the datasets *measured as a percentage of the accuracy of the base decision rule*. The accuracies were surprisingly high, more so with binary data. It is worth pointing out that the accuracy of approximate cumulative dominance in binary datasets ranged from 93.8% to 103.7% of the accuracy of the base decision rule.

In the results discussed so far, approximate dominance was computed by setting $c = 0.99$. This value was selected prior to the analysis based on what this parameter means: $1 - c$ is the expected error rate of the approximation, where *error rate* is the proportion of approximately dominant objects that are not selected by the linear decision rule.

Figure 3, left panel, shows how well the approximation fared in the 51 datasets with various choices of the parameter $c$. The vertical axis shows the mean error rate of the approximation. With numeric data, the error rates were reasonably close to the expected values. With binary data, error rates were substantially lower than expected.

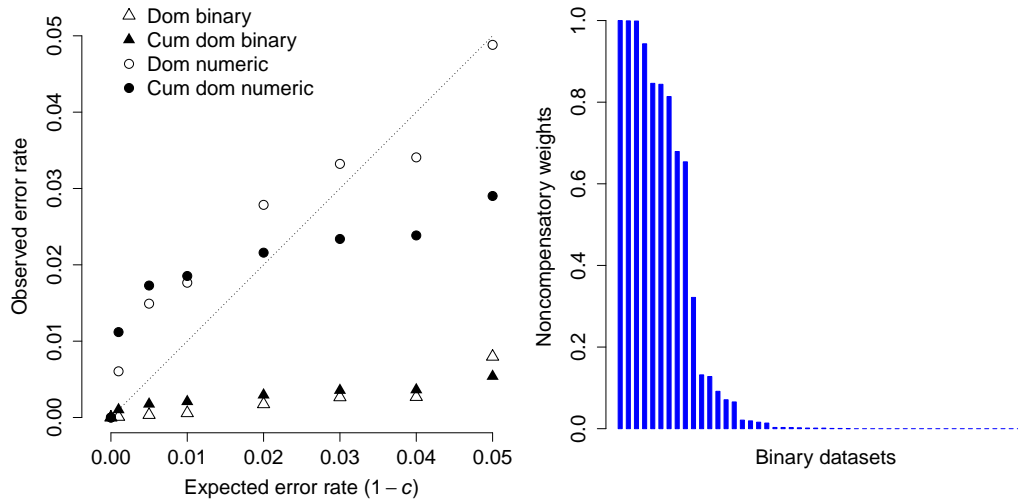

Figure 3: *Left:* Error rates of approximate dominance with various values of the approximation parameter $c$. *Right:* Proportion of linear models with noncompensatory weights in each of the datasets.

**Noncompensatoriness**  Let *noncompensation* be a logical variable that equals TRUE if the decision of the first discriminating cue, when cues are processed in nonincreasing magnitude of the weights, is identical to the decision of the linear decision rule. With binary cues and noncompensatory weights, noncompensation is TRUE with probability 1. Otherwise, its value depends on cue values. If noncompensation is TRUE, the linear decision rule and the corresponding lexicographic rule make identical decisions.

Figure 3, right panel, shows the proportion of base decision rules with noncompensatory weights in binary datasets. Recall that a large number of base decision rules were learned on each dataset, using different training sets and random seeds. The proportion of base decision rules with noncompensatory weights ranged from 0 to 1, with a mean of 0.17 across datasets. Nine datasets had values greater than 0.50. Thirty-two datasets had values less than 0.01.

Figure 4 shows noncompensation in each dataset, together with the accuracies of the base decision rule and the corresponding lexicographic rule. The accuracies on the same dataset are connected by a line segment. The figure reveals overwhelmingly large levels of noncompensation, particularly for binary data. Median noncompensation was 0.85 in numeric datasets and 0.96 in binary datasets. Consequently, the accuracy of the lexicographic rule was very close to that of the linear decision rule: its median accuracy relative to the base decision rule was 96% in numeric datasets and 100% in binary datasets. In summary, although noncompensatory weights were not particularly prevalent in the datasets, actual levels of noncompensation were very high.

## 5   Discussion

It is fair to conclude that all three environmental structures are prevalent in natural environments to such a high degree that decisions guided by these structures approach, and occasionally exceed, the base decision model in predictive accuracy.

We have not examined the performance of any particular decision heuristic, which depends on the cue directions and cue order it uses. These will not necessarily match those of the linear decision rule.[3] The results here show that it is possible for decision heuristics to succeed in natural environments by imitating the decisions of the linear model using less information and less computation—because the conditions that make it possible are prevelant—but not that they necessarily do so.

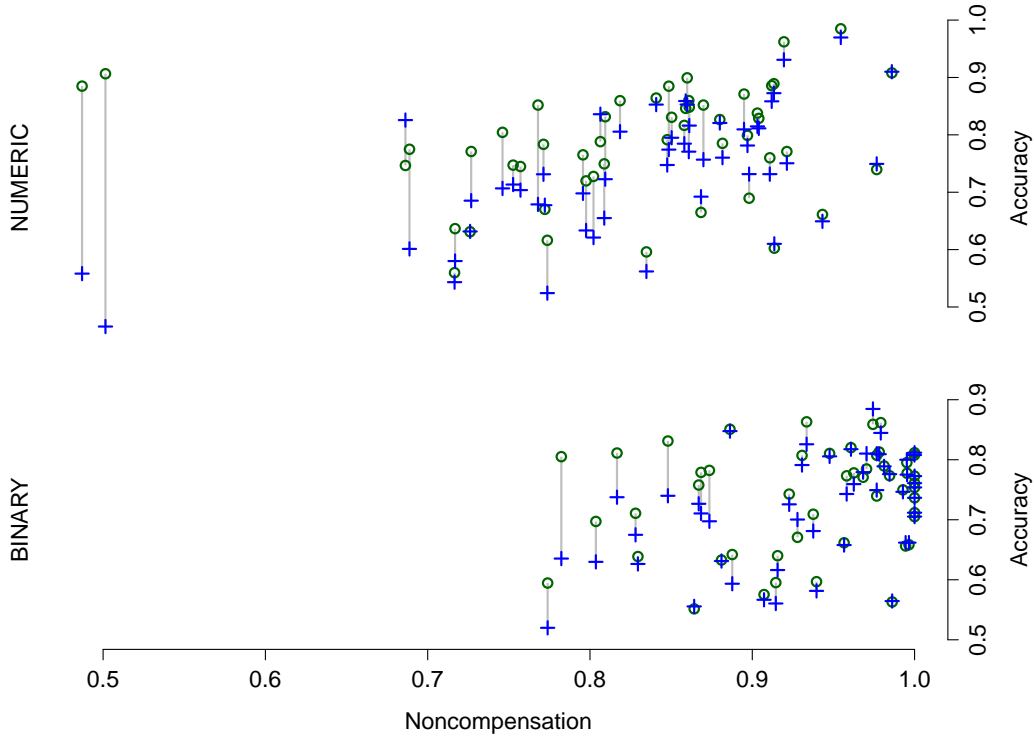

Figure 4: *Prevalence of noncompensation.* For each dataset, the proportion of decisions in which noncompensation took place are plotted against the accuracy of the base decision rule (displayed in green circles) and the accuracy of the corresponding lexicographic rule (displayed in blue plus signs). Accuracies on the same dataset are connected by a line segment.

When decision heuristics are examined through the lens of bias-variance decomposition [23, 24, 25], the three environmental structures examined here are particularly relevant for the bias component of the prediction error. The results presented here suggest that while simple decision heuristics examine a tiny fraction of the set of linear models, in natural environments, they may do so without introducing much additional bias.

It is sometimes argued that the environmental structures discussed here, noncompensatoriness in particular, are relevant for model fitting but not for prediction on unseen data. This is not accurate. The results reviewed in Sections 2.1–2.3 apply to a linear model regardless of how the linear model was trained. If we are comparing objects that were not used to train the model, as we have done here, the discussion pertains to predictive accuracy.

The probabilistic approximations of dominance and of cumulative dominance introduced in this paper can be used as decision heuristics themselves, combined with any method of estimating cue directions and cue order. I leave detailed examination of their performance for future work but note that the results here are encouraging.

Finally, I hope that these results will stimulate further research in statistical properties of decision environments, as well as cognitive models that exploit them, for further insights into higher cognition.

## Acknowledgments

I am grateful to all those who made their datasets available for this study. Thanks to Gerd Gigerenzer, Konstantinos Katsikopoulos, Amit Kothiyal, and three anonymous reviewers for comments on earlier versions of this manuscript, and to Marcus Buckmann for his help in gathering the datasets. This work was supported by Grant SI 1732/1-1 to Özgür Şimşek from the Deutsche Forschungsgemeinschaft (DFG) as part of the priority program "New Frameworks of Rationality" (SPP 1516).

## Footnotes

[1]The authors found 3 datasets in which the weights were noncompensatory and the order of the weights was identical to the cue order of the take-the-best heuristic [19]. It is possible that additional datasets had noncompensatory weights but did not match the take-the-best cue order.

[2]The datasets included the 20 datasets in Czerlinski, Gigerenzer & Goldstein [20], which were used to obtain the two sets of earlier results discussed above [8, 4].

[3]When this is the case, it should be noted, the decision heuristic may have a higher predictive accuracy than the linear model.

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
