[Supplementary Material]

# Linear Decision Rule as Aspiration
# for Simple Decision Heuristics
## SUPPLEMENTARY MATERIAL

**Özgür Şimşek**
Center for Adaptive Behavior and Cognition
Max Planck Institute for Human Development
Lentzeallee 94, 14195 Berlin, Germany
ozgur@mpib-berlin.mpg.de

This document describes the datasets used in the empirical analysis. There were 51 datasets in total, obtained from a wide variety of sources, including online data repositories, textbooks, research publications, packages for R statistical software, and individual scientists collecting field data. The subjects were diverse, including biology, business, computer science, ecology, economics, education, engineering, environmental science, medicine, political science, psychology, sociology, sports, and transportation. The datasets varied in size, ranging from 12 to 601 objects, corresponding to 66–180,300 distinct paired comparisons between the objects. Number of attributes ranged from 3 to 21.

**AFL** OBJECTS: 41 Australian Football League (AFL) games at the Melbourne Cricket Ground in 1993 and 1994 CRITERION: attendance CUES: forecasted maximum temperature on the day of the game, total attendance at other AFL games in Melbourne and Geelong on the day of the game, total membership in the two clubs whose teams were playing, number of players in the top 50 who participated in the game SOURCE: This dataset was assembled by Rowan Todd and Mark McNaughton for a class project at the University of Queensland in a statistics course taught by Dr. Margaret Mackisack. The data sources were *The Football Bible '94* by Rex Hunt, *The Weekend Australian*, *Inside Football*, and *Football Record*. The dataset is available from OzDASL data library [1] under the name AFL Crowd Attendance at the MCG.

**Athletes** OBJECTS: 202 athletes CRITERION: blood hemoglobin concentration CUES: body mass index, sum of skin folds, percent body fat, lean body mass, height, weight, sex, the sport the athlete competes in (basketball, field, gymnastics, netball, rowing, track 400m, swimming, sprint, tennis, water polo) SOURCE: The data were collected by Telford & Cunningham [2] at the Australian Institute of Sport. The dataset is reported in Maindonald & Braun [3] and is available from associated R package DAAG [4] under the name ais.

**Birth weight** OBJECTS: 189 newborns CRITERION: birth weight CUES: age of mother, weight of mother at last menstrual period, race (white, black, other), number of previous premature labors, number of physician visits during the first trimester, presence of uterine irritability, whether the mother smoked during pregnancy, whether the mother has a history of hypertension SOURCE: The data were collected at Baystate Medical Center in Springfield, Massachusetts in 1986 [5]. The dataset is available from R package MASS [6, 7] under the name birthwt.

**Body fat** OBJECTS: 252 males CRITERION: percentage of body fat determined by underwater weighing CUES: age, weight, height, and various body circumference measurements: neck, chest, abdomen, hip, thigh, knee, ankle, biceps, forearm, wrist SOURCE: The data were collected by Penrose et al. [8]. The dataset is available from StatLib [9] under the name bodyfat.

**Cars** OBJECTS: 93 passenger cars on sale in the United States in 1993 CRITERION: sale price of the most basic version of the car CUES: city mileage, highway mileage, number of cylinders, engine size, maximum horsepower, engine revolutions per mile in highest gear, fuel tank capacity, passenger capacity, length, wheelbase, width, weight, rear seat room, luggage capacity, u-turn space, airbag (none, driver only, both driver and passenger), whether a manual transmission version is available, whether the manufacturer is from the United States, type of car (small, sporty, compact, midsize, large, van), drive-train type (rear, front, four-wheel drive) SOURCE: The dataset was assembled by Lock [10] using information from *PACE New Car & Truck 1993 Buying Guide* and *Consumer Reports April 1993 Annual Auto Issue*. It is available from R package MASS [6, 7] under the name Cars93.

**Cigarettes** OBJECTS: 25 brands of cigarettes CRITERION: carbon monoxide emitted from the cigarette smoke CUES: weight, tar content, nicotine content SOURCE: The data were produced by the Federal Trade Commission. The dataset is reported in Mendenhall & Sincich [11]. It is available from the *Journal of Statistics Education* [12].

**Cities** OBJECTS: 76 cities in Germany with more than 100,000 inhabitants CRITERION: population CUES: whether the city has a team in the major soccer league Bundesliga, whether the city is a state capital, whether the city was formerly in East Germany, whether the city is in the industrial belt, whether the abbreviation for the city on license plates is one-letter long, whether the city is on the intercity train line, whether the city hosted a trade fair in 2013, whether the city is the national capital, whether the city has a university SOURCE: This dataset originally appeared in Gigerenzer & Goldstein [13]. It was updated to reflect 2013 population data and attributes. Population data were obtained from the Federal Statistical Office (Das Statistische Bundesamt). They are based on the 2011 census and population density of revision 31.12.2012. The trade fair data are obtained from AUMA, the Association of the German Trade Fair Industry. The dataset reflects only the trade fairs in AUMA category *international and national events*. The industrial belt is the region of Germany known as Ruhrgebiet. The intercity train line includes IC and ICE stops. Universities include *Universität*, *Institut für Technologie*, *Technische Universität*, and *Technische Hochschule*.

**Contraception** OBJECTS: 152 localities in the world (most are United Nations members but includes areas like Hong Kong that are not independent countries) CRITERION: percentage of unmarried women using a modern method of contraception CUES: annual population growth rate, per capita 2001 gross domestic product, percentage of females over age 15 who are economically active, population, expected number of live births per female in 2000, percentage of population that is urban in 2001 SOURCE: The data were collected by the United Nations in 2000–2003. The dataset is reported in Weisberg [14] and is available from associated R package alr3 [15] under the name UN3.

**CPU** OBJECTS: 209 central processing units on the market in 1981–1984 CRITERION: published performance on a benchmark mix (measured relative to a base machine, the IBM 370/158 Model) CUES: cycle time, minimum main memory, maximum main memory, cache memory, minimum number of channels, maximum number of channels SOURCE: The dataset was assembled by Ein-Dor & Feldmesser [16] using information from *Computerworld*. It is available from R package MASS [6, 7] under the name cpus.

**Crime** OBJECTS: 47 states in the United States CRITERION: crime rate in 1960 CUES: percentage of males aged 14–24 in state population, indicator variable for a southern state, mean years of schooling of the population aged 25 years or over, per capita expenditure on police protection in 1960, per capita expenditure on police protection in 1959, labor force participation rate of civilian urban males in the age-group 14–24, number of males per 100 females, state population in 1960, percentage of nonwhites in the population, unemployment rate of urban males 14–24, unemployment rate of urban males 35–39, wealth (median value of transferable assets or family income), income inequality (percentage of families earning below half the median income), probability of imprisonment (ratio of number of commitments to number of offenses), average time served by offenders in state prisons before their first release SOURCE: The data was assembled by Ehrlich [17] from FBI Uniform Crime Reports, U.S. Census, and National Prison Statistics Bulletins. The dataset is available from OzDASL [1] under the name uscrime.

**Diamonds** OBJECTS: 308 round diamond stones CRITERION: sale price CUES: weight in carats, color purity (D, E, F, G, H, I), clarity (internally flawless, very very slight inclusion 1, very very slight

inclusion 2, very slight inclusion 1, very slight inclusion 2), certification (Gemmological Institute of America, International Gemmological Institute, Hoge Raad Voor Diamant)  SOURCE: The dataset was assembled by Chu [18] from advertisements in Singapore's *Business Times* edition of February 18, 2000. It is available from the Journal of Statistics Education [19] and from R package `Ecdat` [20] under the name `Diamond`.

**Dropouts**  OBJECTS: 63 public high schools in Chicago  CRITERION: dropout rate  CUES: enrollment, attendance rate, graduation rate, parental involvement rate, percent limited-English students, percent low-income students, average class size, percent White students, percent Black students, percent Hispanic students, percent Asian students, percent minority teachers, average composite ACT score, IGAP scores: reading, math, science, social science, writing  SOURCE: The data were assembled by Morton [21] and Rodkin [22] from Illinois State Board of Education's 1994 School Report Card and other sources.

**Extramarital affairs**  OBJECTS: 601 married individuals  CRITERION: number of extramarital affairs in the past year  CUES: sex, age, numbers of years in marriage, whether the individual has children, degree of religiosity (on a scale from 1 to 5), level of education, occupation (on a scale from 1 to 7, according to Hollingshead classification), marital happiness (on a scale from 1 to 5)  SOURCE: The dataset was created by Fair [23] using responses to two magazine surveys: by *Psychology Today* in 1969 and by *Redbook* in 1974. It is available from R package `Ecdat` [20] under the name `Fair`.

**Fish**  OBJECTS: 413 female Arctic charr  CRITERION: number of eggs  CUES: age, weight, mean egg weight  SOURCE: The data were collected by Christian Gillet, French National Institute for Agricultural Research. The dataset was obtained via personal communication in 2012.

**Forest fires**  OBJECTS: 517 forest fires in the Montesinho Natural Park, Portugal  CRITERION: burned area  CUES: temperature, relative humidity, wind speed, accumulated precipitation within the previous 30 minutes, month of the year, day of the week, location of the fire: *x*-axis and *y*-axis spatial coordinates within the Montesinho park map, ranging from 1 to 9.  SOURCE: The dataset was assembled by Cortez & Morais [24] from measurements obtained by Bragança Polytechnic Institute and by the inspector responsible for the Montesinho fire occurrences from January 2000 to December 2003. It is available from the UCI Machine Learning Repository [25] under the name `Forest Fires`.

**Fuel consumption**  OBJECTS: 48 contiguous states of the United States  CRITERION: per capita motor fuel consumption in 1972  CUES: population, fuel tax rate, per capita income, miles of federal-aid primary highways, proportion of the population who are licensed drivers  SOURCE: The data were collected by Christopher Bingham for the American Almanac for 1974, except for fuel consumption, which was given in the 1974 World Almanac. The dataset is reported in Weisberg [26]. It is available from StatLib [9], in collection `alr`, under the name `alr35`.

**Galápagos**  OBJECTS: 29 islands in the Galápagos archipelago  CRITERION: number of plant species  CUES: surface area, elevation, distance to the nearest island, surface area of the nearest island, distance from the center of the archipelago  SOURCE: The dataset was assembled by Johnson & Raven [27]. It is reported in Weisberg [14] and is available from associated R package `alr3` [15] under the name `galapagos`. Elevation of six very small islands, which were not recorded in the original dataset, are taken from OzDASL [1] (`Galápagos Island Species Data`).

**Gambling**  OBJECTS: 47 British teenagers  CRITERION: annual gambling expenditure  CUES: sex, socio-economic status, weekly income, verbal score  SOURCE: The data were collected by Ide-Smith & Lea [28]. The dataset is reported in Faraway [29] and is available from associated R package `faraway` [30] under the name `teengamb`.

**Highway**  OBJECTS: 39 segments of highway in Minnesota  CRITERION: accident rate  CUES: segment length, average daily traffic count, truck volume as a percent of total volume, speed limit, number of lanes, lane width, shoulder width, number of signalized interchanges per mile, number of freeway-type interchanges per mile, number of access points per mile, highway type (federal interstate highway, principal arterial highway, major arterial, other)  SOURCE: The data were taken from an unpublished master's paper in civil engineering by Carl Hoffstedt. The dataset is reported in Weisberg [14] and is available from associated R package `alr3` [15], under the name `highway`.

**Hitters**  OBJECTS: 263 hitters in North American Major League Baseball  CRITERION: annual salary at the beginning of the 1987 season  CUES: 1986 performance: number of at bats, hits, home

runs, runs scored, runs batted in, walks, putouts, assists, errors; career performance: number of at bats, hits, home runs, runs scores, runs batted in, walks; number of years in the major leagues, player's division at the end of the 1986 season, player's league at the end of the 1986 season, player's league at the beginning of the 1987 season SOURCE: The dataset was prepared by the Statistical Graphics Section of the American Statistical Association for the 1988 Annual Statistical Meetings. It is available from StatLib [9] `baseball`. The version used here is the dataset `BaseballHitters` from Fox [31, 32], which includes corrections from Hoaglin & Velleman [33].

**Homeless** OBJECTS: 50 cities in the United States CRITERION: rate of homelessness CUES: mean temperature, unemployment rate, percentage of inhabitants with incomes below the poverty line, vacancy rate, population, percentage of public housing, whether the city has rent control SOURCE: The dataset was assembled by Tucker [34].

**Hospitals** OBJECTS: 12 naval hospitals of the United States CRITERION: monthly man-hours associated with maintaining the anesthesiology service CUES: number of surgical cases, eligible population, number of operating rooms SOURCE: The dataset is from Myers [35]. It is reported in Hand et al. [36], with identifying number 269.

**Houses** OBJECTS: 27 houses sold in Erie, Pennsylvania CRITERION: Selling price CUES: Current tax, number of bathrooms, number of rooms, number of bedrooms, number of fireplaces, number of garage spaces, lot size, total living space, age of house, construction type (brick, frame, brick and frame, aluminum and frame), style (ranch, two story, one and a half story) SOURCE: The data are from Narula & Wellington [37]. Partial dataset is reported in Weisberg [26] and is available from StatLib [9], in collection `alr`, under the name `alr241`.

**Ice cream** OBJECTS: 30 four-week periods CRITERION: ice cream consumption per capita CUES: price of ice cream per pint, weekly family income, mean temperature SOURCE: The data are from Hildreth & Lu [38]. The dataset is used in Kadiyala [39] and is reported in Hand et al. [36] with identifying number 268.

**Infant mortality** OBJECTS: 105 nations CRITERION: infant-mortality rate CUES: per-capita income, geographic location (Africa, Americas, Asia, Europe), whether the country exports oil SOURCE: The dataset is reported in Fox [31, 32]. Infant-mortality rates were obtained by Leinhardt & Wasserman [40] from the editorial section of *The New York Times* [41].

**Jets** OBJECTS: 22 jet fighter aircraft of the United States Navy and Air Force CRITERION: First flight date, in months after January 1940 CUES: Specific power, flight range factor, payload as a fraction of gross weight, sustained load factor, whether the aircraft can land on a carrier SOURCE: The data are from Stanley & Miller [42]. The dataset is reported in Hand et al. [36] with identifying number 110.

**Lakes** OBJECTS: 69 world lakes CRITERION: number of known crustacean zooplankton species present in the lake CUES: surface area, maximum depth, mean depth, specific conductance, elevation, latitude, longitude, distance to nearest lake, number of lakes within 20 km, rate of photosynthesis SOURCE: The data are provided by Dodson [43]. The dataset is reported in Weisberg [14] and is available from associated R package `alr3` [15] under the name `lakes`.

**Land rent** OBJECTS: 67 counties in Minnesota, USA CRITERION: rent per acre paid in 1977 for agricultural land planted in alfalfa CUES: average rent for all tillable land, density of dairy cows, proportion of pasture land, whether liming is required to grow alfalfa SOURCE: The data were collected by Douglas Tiffany. The dataset is reported in Weisberg [14] and is available from associated R package `alr3` [15] under the name `landrent`.

**Mammals** OBJECTS: 58 mammal species CRITERION: average daily sleep CUES: body weight, brain weight, maximum life span, gestation time, predation index, sleep exposure index, overall danger index. SOURCE: The data are from a study by Allison & Cicchetti [44]. The dataset is available from StatLib [9] under the name `sleep`.

**Men** OBJECTS: 34 famous men CRITERION: mean attractiveness rating CUES: mean likeability rating, name recognition, whether the man is American SOURCE: The data were collected by Henss [45] with the participation of 115 male and 131 female Germans, aged 17–66 years.

**Mileage** OBJECTS: 398 cars built in 1970–1982 CRITERION: Mileage CUES: number of cylinders, engine displacement, horsepower, vehicle weight, time to accelerate from 0 to 60 mph, model

year, origin (American, European, Japanese) SOURCE: The dataset was prepared by the Committee on Statistical Graphics of the American Statistical Association (ASA) for its Second Exposition of Statistical Graphics Technology, held in conjunction with the Annual Meetings in Toronto, August 15–18, 1983. It is available from StatLib [9] under the name `cars`. The version used in the paper is from the UCI Machine Learning Repository [25] (`Auto+MPG`), in which 8 of the original cars were removed because their mileage values were missing.

**Mines** OBJECTS: 44 coal mines in the Appalachian region of western Virginia CRITERION: number of fractures in upper seams of coal mines CUES: inner burden thickness, percent extraction of the lower previously mined seam, lower seam height, duration of operation SOURCE: The dataset is reported in [46] and is available from associated R package `mpg` [47] under the name `p13.7`.

**Mortality** OBJECTS: 60 metropolitan areas in the United States CRITERION: mortality rate CUES: average annual precipitation, average January temperature, average July temperature, percent population aged 65 or older, average household size, median school years completed by those over 22, percent housing units that are sound and with all facilities, humidity, population density in urbanized areas, percent nonwhite population in urbanized areas, percent employed in white collar occupations, percentage of families with income less than $3000, relative hydrocarbon pollution potential, relative nitric oxides pollution potential, relative sulfur dioxide pollution potential, annual average relative humidity SOURCE: The dataset was assembled by McDonald & Schwing [48]. It is available from StatLib [9] under the name `pollution`.

**Obesity** OBJECTS: 136 children CRITERION: somatotype (a scale from 1, very thin, to 7, obese, of body type) CUES: sex, body measurements at ages 2, 9, and 18: height, weight, leg circumference, strength SOURCE: The data were collected by Tuddenham & Snyder [49] on children born in Berkeley, California, between January 1928 and June 1929. The dataset is reported in Weisberg [14] and is available from associated R package `alr3` under the name `BGSall`.

**Occupation** OBJECTS: 36 occupations CRITERION: prestige rating of the National Opinion Research Center (NORC) CUES: suicide rate among males aged 20–64, median income, median number of school years completed (16+ was treated as 16). SOURCE: The dataset was assembled by Labovitz [50] using data from the U.S. Census of 1950, prestige rankings obtained by NORC in its 1947 survey, and other sources. It is reported in Hand et al. [36] with identifying number 490.

**Oxidant** OBJECTS: 30 summer days in Los Angeles, California CRITERION: maximum level of an oxidant CUES: morning averages of four meteorological variables: wind speed, temperature, humidity, insolation SOURCE: The data were collected by the Los Angeles Pollution Control District. The dataset is reported in Rice [51] (pp. 567–570).

**Ozone** OBJECTS: 13 summers in San Francisco, California CRITERION: Summer quarter maximum hourly average ozone reading in San Francisco CUES: Average winter precipitation in the San Francisco Bay area for the preceding two years, summer quarter maximum hourly average ozone reading at San Jose, year of ozone measurement SOURCE: The data are from a study by Sandberg [52] whose source were measurements provided by the Bay Area Air Pollution Control District. The dataset is reported in Weisberg [26] and is available from StatLib [9], collection `alr`, under the name `alr63`.

**Paramo** OBJECTS: 14 isolated islands of paramo vegetation in the northern Andes CRITERION: number of bird species CUES: area, elevation, distance from Ecuador, distance to nearest island SOURCE: The data are from a study by Vuilleumier [53]. The dataset is reported in Hand et al. [36] with identifying number 52.

**Pinot Noir** OBJECTS: 38 samples of Pinot Noir wine CRITERION: quality CUES: clarity, aroma, body, flavor, oakiness, region SOURCE: The dataset is reported in Montgomery [46] and is available from associated R package `MPV` [47] with label `table.b11`.

**Pitchers** OBJECTS: 176 pitchers in North American Major League Baseball CRITERION: annual salary at the beginning of the 1987 season CUES: 1986 performance: wins, losses, earned run average, game appearances, innings pitched, games saved; career performance: wins, losses, earned run average, game appearances, innings pitched, games saved; years in major leagues; league at the end of 1986 (American, National); league at the beginning of the 1987 season (American, National) SOURCE: The dataset was prepared by the Statistical Graphics Section of the American Statistical Association for the 1988 Annual Statistical Meetings. It is available from online data repository

StatLib under the name `baseball`. The version used here is the dataset `BaseballPitchers` from Fox [31, 32].

**Plasma** OBJECTS: 315 adults CRITERION: Plasma retinol level CUES: age, sex, body mass index, daily caloric intake, daily fat intake, daily fiber intake, daily cholesterol intake, dietary beta-carotene consumed per day, dietary retinol consumed per day, number of alcoholic drinks consumed per week, smoking status (never smoked, former smoker, current smoker), vitamin use (often, used but not often, not used) SOURCE: The dataset was made available at StatLib [9], under the name `Plasma_Retinol`, by Dr. Therese Stukel, Dartmouth Hitchcock Medical Center. A related publication is Nierenberg et al. [54].

**Rainfall** OBJECTS: 24 days in Coral Gables, Florida CRITERION: amount of rainfall CUES: whether the clouds were seeded or not, percent cloud cover, amount of rainfall one hour before seeding, number of days since the first day of the experiment, suitability for seeding, whether the radar echo was moving or stationary SOURCE: The data were collected by Woodley et al. [55]. The dataset is reported in Weisberg [14] and is available from associated R package `alr3` under the name `cloud`.

**Rebellion** OBJECTS: 32 Romanian counties in 1907 CRITERION: proportion of villages in which rebellious events took place in the Romanian peasant rebellion of 1907 CUES: proportion of arable land devoted to wheat, proportion of rural population that is illiterate, proportion of land owned in units of 7 to 50 hectares, Gini coefficient of inequality of landownership, population, region (Northern, South Central, Southwest, Eastern) SOURCE: The dataset was assembled by Chirot & Ragin [56]. Partial dataset is reported in Fox [31] under the name `Chirot`.

**Salary** OBJECTS: 52 professors at a Midwestern college in the United States CRITERION: academic year salary CUES: sex, rank (assistant professor, associate professor, full professor), number of years in current rank, the highest degree earned (doctorate, masters), number of years since highest degree was earned SOURCE: The dataset is reported in Weisberg [14] and is available from associated R package `alr3` under the name `salary`.

**Sulfur dioxide** OBJECTS: 41 cities in the United States CRITERION: annual mean concentration of sulfur dioxide CUES: average annual temperature, number of manufacturing enterprises employing 20 or more workers, population, average annual wind speed, average annual rainfall, average number of days with rainfall per year SOURCE: The data were collated by Sokal & Rohlf [57] from several US government publications. The dataset is reported in Hand et al. [36] with identifying number 26.

**Sperm** OBJECTS: 24 heterosexual couples CRITERION: mean sperm count per copulation CUES: age, height, and weight of each of the partners involved, volume of one male teste SOURCE: The data were collected by Baker & Bellis [58]. The dataset is reported in Wood [59] and is available from associated R package `gamair` under the name `sperm.comp2`.

**Tips** OBJECTS: 244 parties dining in a restaurant CRITERION: tip rate CUES: dollar amount of the bill, size of the party, sex of the bill payer, day of the week, time of the day, whether there were smokers in the party SOURCE: Data were recorded by a food server in a restaurant located in a suburban shopping mall in the United States during an interval of two and a half months in early 1990. The dataset was reported in a collection of case studies for business statistics [60]. It is available from R package `reshape` under the name `Tips`.

**Votes** OBJECTS: 159 counties in Georgia, USA CRITERION: proportion of uncounted votes in the 2000 presidential election CUES: type of voting equipment used (optical scan with central count, optical scan with precinct count, punch card, lever, paper), whether the county is in Atlanta, whether the county is urban or rural, the proportion of African Americans in the county, the economic status of the county (rich, middle, poor) SOURCE: The dataset was assembled by Meyer [61]. It is reported in Faraway [29] and is available from associated R package `faraway` [30] under the name `gavote`.

**Waste** OBJECTS: 20 days of a laboratory experiment CRITERION: oxygen absorbed by dairy waste kept in suspension in water CUES: biological oxygen demand, chemical oxygen demand, total Kjedahl nitrogen, total solids, total volatile solids SOURCE: The data are from an experiment by Moore [62]. The dataset is reported in Weisberg [14] and is available from associated R package `alr3` under the name `dwaste`.

**Wheat** OBJECTS: 24 samples of ground wheat CRITERION: protein content measured by the standard Kjeldahl method CUES: measurements of the reflectance of near-infrared radiation at six different wavelengths in the range 1680–2310 nm SOURCE: The data are from a study by Fearn [63]. The dataset is reported in Hand et al. [36] with identifying number 509.

**Women** OBJECTS: 30 famous women CRITERION: mean attractiveness rating CUES: mean likability rating, name recognition, whether the woman is American SOURCE: The data were collected by Henss [45] with the participation of 115 male and 131 female Germans, aged 17–66 years.