[Reviews · NeurIPS 2013]

Submitted by Assigned_Reviewer_1

This work turns NIPS on its head: In NIPS, linear threshold classifiers are
usually considered to be a baseline model, but in this work, linear threshold
classifiers are considered too sophisticated to characterize human decision
making, and various simpler alternatives are explored. The work appears to
make two important contributions: (1) to show that for natural environments,
various decision-making short cuts perform well, and (2) to introduce novel
decision short cuts that are probabilistic generalizations of previously
proposed short cuts.

The paper is very well written and accessible. It should have
more discussion of the implications of the results. The authors should also
make their theoretical contribution clearer. Is approximate (cumulative)
dominance is new to this work? The motivation for the approximation might be
made clearer as well. I believe the reason is that the conditions in section
2 often do not apply, but an approximate rule would be more often applicable,
and as long as it is fairly accurate, it should boost performance. This
appears to be the case in the results.

One additional point about the exposition: The introduction fails to give all the background necessary to motivate the work to a typical NIPS attendee. Before discussing fast-and-frugal heuristics, the authors should discuss the empirical literature showing that simple linear-threshold classifiers are better than
human decision makers (e.g., http://www.sciencemag.org/content/243/4899/1668). If you start from this perspective, it motivates the treatment of the linear-threshold
classifier as the target, and the question of what heuristics approximate this target. Without this background for the reader, I suspect the typical NIPS attendee will wonder the most sophisticated model of decision making is a linear-threshold classifier.

The work claims to be a serious analysis of structure of natural environments,
but I'm far from clear whether the 51 real-world data sets involve the type of
decision making tasks that humans might typically make. It might be worth
moving Table 1 to the supplementary materials (along with the page of
references), and instead to present an argument for why these data sets are
representative of human decision making tasks.

I was hoping the authors would compare their 51 natural data sets to randomly
generated data sets, where the weights and features are drawn from specified
distributions. It would also be interesting to determine the distributional
assumptions that lead to success of the various decision making short cuts.

I am confused about the trade off in setting the cutoff value for the
approximate dominance schemes: A high cutoff means that the
approximate-dominance relationship predicts the correct decision, but also
that fewer approximate-dominance relationships will hold between pairs. The
trade off is not discussed. Was the chosen cutoff (.99) arbitrary, or are the
results sensitive to this choice?
Summary: This work makes a theoretical contribution -- suggesting several novel decision-making short cuts -- and an empirical contribution -- exploring the characteristics of 51 problem domains (I wouldn't go so far as to call them natural human decision-making environments).

Submitted by Assigned_Reviewer_6

The author obtain 51 datasets on which they apply linear decision rules for classification and three variants thereof that have been suggested as capturing the notion of heuristic in that they allow computing a decision variable with less input compared to the full feature values. The authors essentially quantify how well these previously proposed ‘heuristics’ give a classification performance close to the one of their sleected decision criterion applied to regularized linear regression.

This is a nice study on a topic quite relevant for the cognitive science community, but it just feels utterly displaced at NIPS. While I think that the results communicated herein are interesting they should be submitted to a journal such as ‘cognition’. Mehtodologically, the study does not provide anything new and there are several question about why the specific methods were chosen that the text does not answer.

072: Did you want to say that all $sgn(w_i)\Deltax_i$ are nonnegative or that all $sgn(w_i)$ and all $\Deltax_i$ are nonnegative?

111: I cannot understand the sentence: ‘We can not say with certainty…’. The linear rule will use $\Sum_i(w_i \Delta x_i)$ so that arguments about the number of terms being positive or negative alone seem weak.

126: What do you mean by stating that dominance and noncompensatoriness make paired comparisons easier? Computationally? For the brain? Does this include the additional cost for deciding when to use and when not to use it?

139: Why did you use a linear regression? Why not use logistic regression?

140: What prior does the used regularization correspond to? Are those priors appropriate for the given datasets?
Summary: Analysis of classification performance on a group of datasets comparing one specific decision rule based on linear regression with elastic net regularization to previously proposed simplified decision rules regarded as ‘heuristics’. The authors show that the simplified features work somehow well in many cases which they interpret as evidence for the viability of these heuristics.

Submitted by Assigned_Reviewer_7

In this paper the authors investigate the extent to which linear decision rules can be approximated by simple heuristics across a range of 51 “natural” data sets. They do this in two ways. First they train a set of linear models for each of these 51 problems and examine the properties of the weights that can guarantee the success of certain heuristics. In particular they look at the extent to which the learned linear decision rules show the properties of dominance, cumulative dominance and noncompensatoriness. Separately, they also consider the actual performance of the heuristics to assess their performance when the respective properties (e.g. dominance) are not present. Intriguingly, they find that many of the learned linear rules show all of these properties and the performance of the heuristic decision rules is high. This suggests that natural problems are structured such that simple heuristics are effective.

I really enjoyed reading this paper, it is beautifully written and very accessible – for instance, how the introduce the three properties shows perfect balance of rigor and clarity. I also think that the results are intriguing and important. From my own perspective in neuroscience, the idea of trying to enumerate the statistics of natural decisions reminds me of the work on natural scene statistics in vision. This work was hugely important for building our understanding of visual areas of the brain and I wonder whether this kind of analysis will lead to similar insights for higher cognition.

My main complaint with this paper is that the discussion is too short. While I’m not looking for pages and pages in a NIPS paper, I think it would be appropriate to talk more about the implications of these findings. I suspect they are motivated to use these heuristics for AI but as stated above, I think there are implications for neuroscience and psychology that would be worth discussing and would make the paper accessible to a wider audience.
Summary: A well written paper that looks at the statistics of "natural" decisions to show why and how simple heuristics can be useful.
Author Feedback

Author rebuttal: Thanks to the reviewers for their thoughtful comments and their appreciation of the paper. Their suggestions are highly appreciated and duly noted. In particular, we will make sure to expand the discussion section, as suggested by both Reviewers 1 and 7.

Reviewer 1:
Yes, the approximations to dominance and to cumulative dominance are new to this paper. And, the reviewer's understanding of the motivation for the approximation is correct. We will work on further clarifying the parts of the paper that introduces the approximation.

The probability cutoff for the approximation (0.99) was selected before the analysis, based on its interpretation in Section 3. Figure 3 (right panel) analyzes the impact of this choice to the extent we found possible given the page limit. If space permits, we can include a bit more on the trade-off involved in the approximation.

We will follow the reviewer's suggestion and move Table 1 to the supplementary materials. We will include more information on each of the datasets, including the decision criterion and the attributes available. In the paper, we will include detailed information on one (or a few) of the datasets.

Reviewer 6:
Reviewer 6 wrote "This is a nice study on a topic quite relevant for the cognitive science community, but it just feels utterly displaced at NIPS." We are surprised that a reviewer found the paper displaced at NIPS. As a paper on how statistical distributions in the natural world allow computational shortcuts, we do believe that it will be of interest to the NIPS community. Note also that NIPS has been a venue for simple decision heuristics in the past, including an invited talk by Gigerenzer (Fast and Frugal Heuristics: The Adaptive Toolbox, NIPS 2004) and a workshop by Todd & Martignon (Workshop on simple inference heuristics vs. complex decision machines, NIPS 1998).

What we mean by "dominance makes paired comparisons easier" is that decisions identical to those of the linear model may be made using any one of the attributes by itself. The reviewer is right, our phrasing was sloppy. We will fix it.

We used linear regression for continuity with the literature. We do not have a reason to believe that the results would be different with logistic regression. In terms of their accuracy on the 51 datasets, linear and logistic regression are comparable: Mean accuracy for both methods (without regularization) is 0.78 in numeric datasets, and 0.74 in binary versions.

The regularization method we used (elastic net) is a standard (and state-of-the art) method (it combines the L1 and L2 penalties). We selected regularization parameters using cross-validation on the training data, which is also a standard approach. We will be happy to try other methods for learning a linear decision rule and are open to specific recommendations.

Reviewer 7:
Thanks for alerting us to the connection to natural scene statistics in vision, we will incorporate it in the paper.